



# A compact static birefringent interferometer for the measurement of upper atmospheric winds: concept, design and lab performance

Tingyu Yan[1,2], Jeffery A. Langille[2], William E. Ward[2], William A. Gault[†], Alan Scott[3], Andrew Bell[5], Driss Touahri[4], Sheng-Hai Zheng[3], and Chunmin Zhang[1]

[1]Institute of Space Optics, Xi'an Jiaotong University, Xi'an, Shaanxi, 710049, China
[2]University of New Brunswick, Fredericton, New Brunswick, E3B 5A3, Canada
[3]Honeywell, Ottawa, Ontario
[4]VIAVI Solutions, Ottawa, Ontario
[5]TNO Space Systems, Stieltjesweg, Netherlands
[†]Deceased

**Correspondence:** William E. Ward (wward@unb.ca), Jeffery Langille (jeff.langille@unb.ca)

**Abstract.** A new compact static birefringent Doppler wind imaging interferometer has been developed for the purpose of observing upper atmospheric winds using suitably isolated airglow emissions. The instrument, called the Birefringent Doppler Wind imaging Interferometer (BIDWIN), combines a field widened birefringent delay plate placed between two crossed Wollaston prisms with an imaging system, waveplates and polarizers to produce four fixed 90-degree phase stepped images of the interference fringes conjugate to the scene of interest. A four-point algorithm is used to extract line of sight Doppler wind measurements across the image of the scene. The arrangement provides a similar throughput to that of a field widened Michelson interferometer, albeit constructed without moving parts. Consequently, the instrument provides a compact, lightweight and robust alternative. In this paper, the instrument concept is presented and the design and optimization of a prototype version of the instrument is discussed. Characterization of the lab prototype is presented and the performance of the instrument is examined by applying the instrument to measure a low velocity two-dimensional Doppler wind field with a high precision (5 m/s) in the lab.

## 1 Introduction

Upper atmospheric motions in the mesosphere and lower thermosphere (MLT) region are dominated by large scale tides, planetary waves, as well as, large-scale and small-scale gravity waves. Indeed, measurements from space-borne platforms were critical to showing that these waves drive the large-scale circulation in the middle atmosphere (Ern et al., 2016; Geller et al., 2013); however, the processes that govern energy dissipation and interaction remain incompletely understood (Fritts et al., 2016). The MLT region is coupled to the upper atmosphere by wave processes that influence the neutral wind field and subsequently impact ionospheric dynamics. Therefore, understanding this coupling and the mechanisms that influence the





dissipation of energy associated with small-scale variability requires the simultaneous spatial sampling of several components

of the dynamical fields at resolutions and uncertainties that allow these processes to be resolved.

Passive measurements of Earth's naturally emitted airglow have been used for several decades to remotely measure upper atmospheric motions. Geophysical variability in the region due to the presence of gravity waves and other motions (tides, planetary waves etc.), perturbs the airglow layer, resulting in variations in the line of sight (LOS) Doppler wind and irradiance field (Hines and Tarasick, 1987, 1993). This paper describes the development of a new type of instrument designed to detect

these variations. The instrument, called, the BIrefringent Doppler Wind imaging INterferometer (BIDWIN), is a compact high-resolution, large throughput interferometer constructed with no moving parts.

Several other interferometric techniques have been developed over the past 50 years for the purpose of detecting upper atmospheric motions using airglow emissions. These instruments have provided valuable insights regarding the dynamics occurring in the region. For example, field widened Michelson interferometers, such as the wide-angle Michelson Doppler imaging in-

terferometer (WAMDII) (Shepherd et al., 1985), the Wind Imaging Interferometer (WINDII) which flew on NASA's Upper Atmosphere Research Satellite (UARS) satellite from 1991 to 2005 (Shepherd, 2002; Shepherd et al., 2012), the mesospheric imaging Michelson interferometer (MIMI) in which a fixed sectored mirror was implemented (Babcock, 2006), as well as the ground based Michelson Interferometer for Airglow Dynamics Imaging (MIADI) (Langille et al., 2013b) and the E-Region Wind Interferometer (ERWIN) (Gault et al., 1996a; Kristoffersen et al., 2013) have been implemented to measure upper at-

mospheric motions using airglow emissions. The Fabry-Perot interferometer and Doppler Asymmetric Spatial Heterodyne (DASH) interferometer have also been implemented to measure upper atmospheric winds, primarly in the thermosphere region. (Hays et al., 1993; Killeen et al., 1999; Anderson et al., 2012; Aruliah et al., 2010; Shiokawa et al., 2012; Englert et al., 2007; Harlander et al., 2010).

Making advancements to the field of interferometric wind measurements requires the development of instruments that

achieve a similar or better accuracy to what is currently possible but with a higher spatial and temporal resolutions using a more robust and less complicated instrument. The core component of the BIDWIN instrument is the field widened birefringent interferometer (Langille et al., 2013a, 2020) placed between two crossed Wollaston prisms. This configuration produces four images of the scene conjugate to the interference fringes at the detector (see Fig. 3). Appropriate placement of waveplates and polarizers in the system produce four 90-degree phase stepping images of the interference fringes. The samples are processed

using fringe analysis algorithms similar to those used in Doppler Michelson Interferometry (DMI) to extract LOS winds. A similar birefringent interferometer has been implemented to measure the high-speed motion of plasma in the H1-Heliac at the Australian National University (ANU) (Howard, 2006). However, the capacity of the system to measure low velocity wind fields (with a precision on the order of <5 m/s) was not investigated. The BIDWIN system is the first application to implement the field widened birefringent polarization interferometer to observations of upper atmospheric winds using low intensity

airglow emissions.

The paper is organized as follows. First, we present the overall requirements, which guide the design of a general high resolution two-beam interferometer capable of wind measurements with precisions <5 m/s. These requirements form the basic specifications that drive the design of the BIDWIN instrument. Second, we present the BIDWIN measurement principles and



highlight the sensitivity of the technique in comparison to the field widened Michelson interferometer. Third, the design and optimization of the instrument is presented and the overall sensitivity to wind measurements is examined using simulated ground-based measurements. Fourth, the implementation, characterization and testing of the instrument is presented. Finally, we examine the performance of the design by performing measurements of low velocity winds produced in the lab.

## 2 Science requirements

### 2.1 Airglow emissions

The Earth's airglow is naturally emitted in the ultra-violet visible and near-visible spectral regions. The choice of airglow emission lines that can serve as useful tracers for Doppler wind measurements is rather limited. However, measurements have been made both from the ground and from satellite and several similar instruments are being considered for future ground stations and space missions. Some of these instruments and missions are summarized in Table 1. The list is not exhaustive but is a good summary of the airglow emissions that have been used, or are planned to be used, in wind measurements. All of these instruments are wide-field Michelson interferometers except for CLIO (Wang et al., 1993), HRDI (Hays et al., 1993) and TIDI (Killeen et al., 1999), which are Fabry-Perot interferometers and MIGHTI which is a Doppler Asymmetric Spatial Heterodyne Spectrometer (DASH) (Englert et al., 2007). The emissions consist of $O^1S$ (oxygen green line, 557.7 nm), $O^1D$ (oxygen red line, 630.0 nm), various bands of the Meinel OH system and the $^1\Sigma$ and $^1\Delta$ band systems of $O_2$. The $O^+$ lines observed by WINDII yielded little useful data, though there might be potential there for more work.

Limb-viewing satellite instruments such as MIGHTI (Englert et al., 2017), WINDII (Shepherd et al., 1993), HRDI, and TIDI can generate altitude profiles of the wind and are capable of providing high spatial sampling and global coverage. Both nightime and daytime measurements are possible from a satellite; however, this is not true of measurements made from the ground. Ground-based measurements can only assign a wind to an assumed typical altitude region for the emission being observed and such measurements are only possible at night. If a satellite instrument is properly baffled to protect the optics from scattered sunlight, observations are possible during both day and night. The oxygen emissions (O and $O_2$) are generally brighter during daytime than night-time.

Characteristics of the different lines are given in Table 2 for satellite measurements and in Table 3 for ground-based measurements. The $O_2\Sigma(0,0)$ and $O_2^1\Delta(0,0)$ bands are too strongly self-absorbed to be useful for ground-based measurements. The values in Table 2 for $O^1S$ and $O_2^1\Delta$ are from Ward et al. (2001), who refer back to (Gault et al., 1996b) for $O^1S$ and to (Thomas et al., 1984) and (Howell et al., 1990) for $O_2^1\Delta$. All of the lines listed in Table 2 apart from the molecular oxygen lines provide similar signal levels; however, the sensitivity of the wind measurements made with a two-beam interferometer is also dependent on the line shape, as well as, the maximum optical path difference of the interferometer. Therefore, we briefly examine the general principle of wind measurements with a two-beam interferometer and examine the sensitivity of wind measurements made using the lines listed in Table 2.

**Table 1.** Projects measuring Earth's upper atmospheric winds remotely using Doppler shifts of airglow emissions.

| Project name | Comments | Emissions observed |
|---|---|---|
| WINDII | UARS[1] satellite, launched 1991 | $O^1S$, $O^1D$, OH(8,3), $O^+$, $O_2{}^1\Sigma(0,0)$ |
| HRDI | UARS satellite, launched 1991 | $O_2{}^1\Sigma$(A, B and $\gamma$ bands) |
| ERWIN | Ground, Resolute/Eureka | $O^1S$, OH(6,2), $O_2{}^1\Sigma(0,1)$ |
| TIDI | TIMED[2] satellite | $O^1S$, $O_2{}^1\Sigma(0,0)$ |
| CLIO | Ground, Resolute | $O^1S$, $O^1D$, OH(7,3) |
| MICADO | Ground, OHP[3] France | $O^1S$, $O^1D$ |
| EPIS | Ground, Spitzbergen, Svalbard | $O^1S$, $O^1D$ |
| Waves | Satellite proposal, not selected | $O^1S$, $O^1D$, $O_2{}^1\Delta(0,0)$ |
| MIADI | Ground, UNB[4] | $O^1S$, $O^1D$, OH(6,2), OH(7,3), $O_2{}^1\Sigma(0,1)$ |
| WaMI | Studies for satellite proposal, UNB | $O^1S$, OH(8,5), $O_2{}^1\Delta(0,0)$ |
| DynAMO | Studies for Mars mission proposal, UNB | $O_2{}^1\Delta(0,0)$ |
| MIGHTI | ICON satellite | $O^1S$, $O^1D$ |

[1] Upper Atmosphere Research Satellite (NASA).

[2] Thermosphere Ionosphere Mesosphere Energetics and Dynamics satellite.

[3] Observatoire de Haute-Provence.

[4] University of New Brunswick.

## 2.2 Wind measurements using a two-beam interferometer

Measurement of Doppler shifts in spectrally isolated airglow emissions using two beam interferometers is achieved by sampling the interference pattern produced with an interferometer at a number of phase steps spanning a full fringe around some large fixed effective path difference. The general measurement process, retrieval algorithms and analytic expressions for the sensitivity of such measurements is described in detail by Kristoffersen et al. (2021). The instrument discussed in this paper samples the interferogram at roughly 90 degrees phase steps. In this case, the intensity of the observed signal can be written as

$$I_i = I_0 \left[ 1 + UV \cos\left( \Phi + \varphi_i \right) \right] \tag{1}$$

where $I_0$ is the mean intensity, $U$ is the instrument visibility, $V$ is the line visibility and $\varphi_i$ is the $i$th phase step. Motion of the source along the line of sight with velocity $w$ results in a slight phase shift in the interferogram given by

$$\delta\Phi = \frac{2\pi D}{c\lambda} w \tag{2}$$

where $\lambda$ is the target wavelength, $D$ is the effective path difference and $c$ is the speed of light. The signal $S$ at the detector is given by

$$S = \frac{10^6}{4\pi} E_0 A\Omega\tau\eta t \tag{3}$$





**Table 2.** Emissions for Earth satellite (limb) observations.

| Emission | Wavelength (nm) | Typical emission temp. (K) | Day/Night | Tangent height (km) | Limb intensity (kR) |
|---|---|---|---|---|---|
| O$^1$S (green line) | 557.7 | 1000 | D | 200 | 20 |
| | | | | 150 | 55 |
| | | | | 100 | 90 |
| | | 200 | N | 110 | 1.2 |
| | | | | 95 | 6.5 |
| O$^1$D (red line) | 630.0 | 1000 | D | / | 30 |
| | | | N | / | 5 |
| O$_2$$^1\Sigma$ | 763.2 | 200 | D | 94 | 200 |
| | | | N | 94 | 20 |
| OH(6,2) P$_1$(3) | 843.1 | 200 | N | 85 | 15 |
| OH(7,3) P$_1$(3) | 892.2 | 200 | N | 85 | 22 |
| O$_2$$^1\Delta$(0,0) | 1264 (strong) | 250 | D | 80 | 640 |
| | | | | 65 | 2700 |
| | 1278 (weak) | 250 | D | 50 | 2300 |
| | | | | 45 | 3000 |
| | 1264 | 200 | N | 85 | 80 |

**Table 3.** Emissions for ground based (night) observations.

| Emission | Wavelength (nm) | Typical emission temp. (K) | Line zenith intensity (R) |
|---|---|---|---|
| O$^1$S (green line) | 558 | 200 | 250 |
| O$^1$D (red line) | 630 | 1000 | 100 |
| O$_2$$^1\Sigma$(0,1) | 866 | 200 | 25 |
| OH(6,2) P$_1$(3) | 843 | 200 | 220 |
| OH(7,3) P$_1$(3) | 892 | 200 | 320 |





where $E_0$ is the average emission rate in Rayleighs, $A$ is the collecting area (cm$^2$), $\Omega$ is the solid angle (sr) at the location of $A$, $\tau$ is the transmittance, $\eta$ is the quantum efficiency of the detector and $t$ is the integration time (s). The product $A\Omega$, called the étendue, is determined by the geometry of the optics. To achieve as large a signal as possible, the instrument should be designed so the product $A\Omega$ is as large as possible, within whatever restrictions exist. The usual way to send light through an interferometer is to place a telescope in front that defines the field of view and passes a well-defined beam into the interference optics, with an image of the entrance aperture half-way through the interferometer. Placing the aperture image nearer one end of the interferometer than the other would restrict the $\Omega$ that could be achieved.

Expressions for the uncertainty in the wind measurement were originally developed for the Michelson by Ward (1988) and by Rochon (2001). Ward tested the expression against a computer model that added noise to the signal levels using a gaussian random number generator. General expressions for the sensitivity of Doppler wind measurements are presented by Kristoffersen et al. (2021). In the ideal case, where four samples are obtained with 90-degree phase steps the expression for the standard deviation, $\sigma_w$, of the wind measurement is

$$\sigma_w = \frac{c\lambda}{\sqrt{2}\pi(\text{SNR})UVD} \tag{4}$$

In Eq. (4), the line visibility is related to the source parameters and the effective path difference of the interferometer as $V = \mathrm{e}^{-QTD^2}$, where for the O$^1$D emission at 630 nm , $Q = 2.87 \times 10^{-5}(\text{cm}^{-2}\text{K}^{-1})$. The instrument visibility is maximized by using crystals with high optical quality and is assumed to be $U \sim 0.99$.

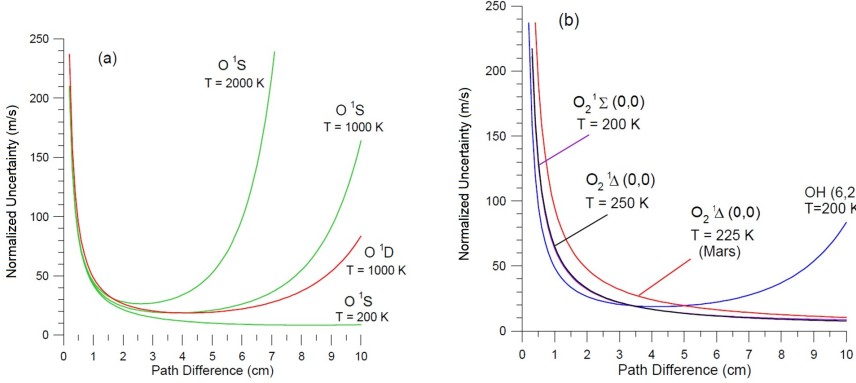

**Figure 1.** Normalized uncertainty plotted against $D$ (0 to 10cm) for several emissions. The O$^1$S emission is shown in green for three temperatures in (a).

For a selected emission, assuming a fixed emission rate and temperature, an optimum path difference exists for the measurement of wind. In Eq. (4), as $D$ increases, $V$ decreases and the graph of $\sigma_w$ vs. $D$ passes through a minimum. Fig. 1 shows plots of the "normalized uncertainty", i.e., the standard deviation of the wind measurement for SNR = 1, plotted as a function of $D$. The curves for O$_2$ have minima beyond $D = 10$ cm. Three curves are shown for O$^1$S, corresponding to three temperatures. $T = 200$ K is typical for the MLT region and 1000 K and 2000 K correspond to the middle thermosphere. Fig. 2 has the same curves





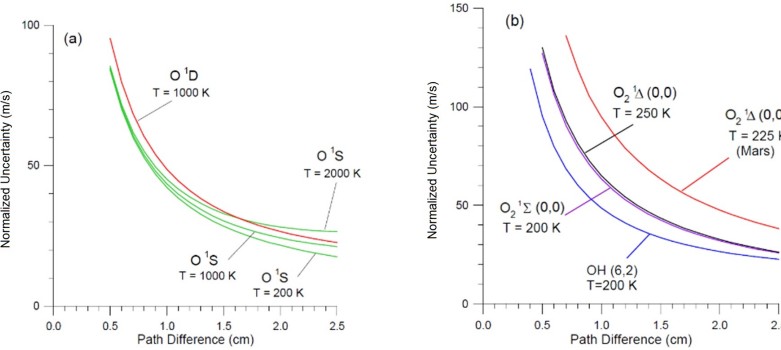

**Figure 2.** Normalized uncertainty plotted against $D$ (0 to 2.5cm) for several emissions. The $O^1S$ emission is shown in green for three temperatures in (a).

as Fig. 1 but shows the $D = 0$ to 2.5 cm region in more detail. For a specific emission, assuming a fixed emission rate and
temperature. The ideal design of a two-beam interferometer is optimized to have an effective path difference near the minima
for a particular emission line. In practice, this may not be possible due to the physical design constraints of the instrument. In
this case, the SNR must be increased to offset the deviation away from optimal configuration.

## 3 The birefringent imaging Doppler wind interferometer (BIDWIN)

The optical layout of BIDWIN is depicted in Fig. 3, where the input is collimated light from the scene of interest. This light is
incident on the first Wollaston prism, the aperture of which defines the entrance aperture of the optical system. The Wollaston
splits the incoming radiation into two orthogonally polarized beams (vertical and horizontal). The objective lens located directly
following the Wollaston forms orthogonally polarized images of the scene in the top frame and bottom frames of the split field
polarizer conjugate to the field stop location. The polarization axes of the two sectors of the split field polarizer are oriented
along the $y$ and $x$ axes corresponding to the orientation of the orthogonal polarizations produced by the Wollaston prisms. A
quarter waveplate is attached to the bottom sector directly behind the polarizer with its optical axis oriented at $45°$ to the $x$
axis. Therefore, the light exiting from the top sector is linearly polarized while the bottom sector is circularly polarized. An
image of the field stop is passed through the field widened delay plate as collimated light by the collimating lens. The delay
plate introduces an optical path difference between the beams where the direction through the plate is mapped to position in
the scene. The second Wollaston prism is positioned behind the field widened delay plate and is rotated $90°$ relative to the
first Wollaston prism. After passage through this prism, the beam is split horizontally and is orthogonally polarized. The light
exiting the Wollaston is collected by the imaging system to produce four images at the detector array of the scene which is
conjugate to the interference fringes of equal inclination. The configuration produces four samples of the interference fringes,
phase stepped by $\pi/2$ as idealized in Fig. 3 assuming a 10 degree field of view.





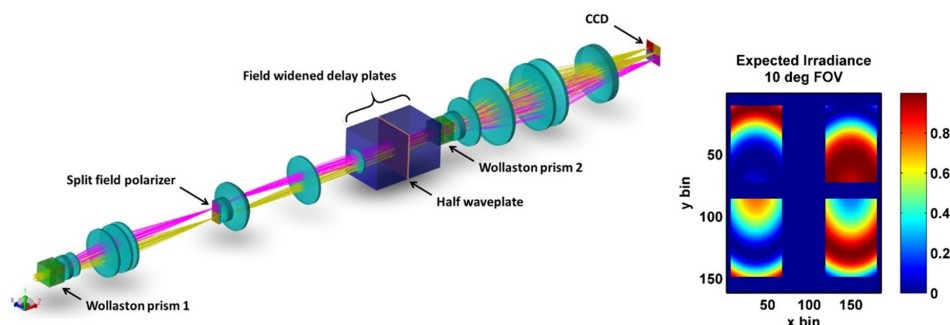

**Figure 3.** The prototype BIDWIN optical layout and simulated interference fringes at the detector.

The field widened delay plate is constructed from two crossed equal length uniaxial birefringent crystal slabs cut with the

optical axis in the plane of the clear aperture with a half waveplate placed between them. The optical axis of the first slab is oriented at $45°$ to the $x$ axis and the second slab is $135°$ to the $x$ axis. The half waveplate is oriented with its optical axis along the $y$ axis to ensure the polarization of the light incident on the second slab and the exiting light from the first slab are symmetric about the $y$ axis. The optical path difference between the extraordinary and ordinary rays through the crystal depends on the incident angle $\theta$ as well as the azimuth $\phi$ of the incident light. In the case that the two slabs and the half waveplate are perfectly

aligned, the optical path difference across the field of view is given to fourth order by Title and Rosenberg (1979)

$$\Delta(\theta,\phi) \approx \Delta_0 \left[ 1 - \frac{1}{4n_o^2}\left(\frac{n_e-n_o}{n_e}\right)\sin^2(\theta) + \frac{\sin^4(\theta)}{8n_o^4} + \frac{1}{8n_o^4}\left(\frac{n_e-n_o}{n_e}\right)\sin^4(\theta)\sin^2(2\phi) \right] \tag{5}$$

where $\Delta_0 = l(n_e - n_o)$, $l$ is the total length of the two slabs. For most birefringent materials, the third term on the right-hand side is extremely small and the azimuthal dependence is negligible. In this case, the device is field-widened, and the optical path difference varies slowly with incident angle. This configuration is extremely sensitive to misalignments or mismatches

between the components and design errors. Exploring these sensitivities has been carried out using a Jones matrix framework that neglects Fresnel effects and Fabry Perot fringes, but takes into account birefringent splitting and unwanted coupling between e and o waves at the interferaces (Langille et al., 2020). In section 4, we use this framework to examine the field of view sensitivity of the optimized design and compare modelled results to lab measurements. Here we use the Jones matrix approach to present the measurement principle.

The incident light of the airglow can be regarded as unpolarized. Therefore, it is split into two orthogonally polarized beams by the first Wollaston prism. As a result, the top beam at the split field polarizer is vertically polarized and can be represented by the Jones vector $\mathbf{E_t} = 1/\sqrt{2}\left[\begin{smallmatrix}0\\1\end{smallmatrix}\right]$, while the bottom beam is horizontally polarized and can be represented by the Jones vector $\mathbf{E_b} = 1/\sqrt{2}\left[\begin{smallmatrix}1\\0\end{smallmatrix}\right]$. For a perfectly aligned split field polarizer, the Jones matrix of the top polarizer is $\mathbf{J_t} = \left[\begin{smallmatrix}0&0\\0&1\end{smallmatrix}\right]$, while the Jones matrix of the bottom polarizer is $\mathbf{J_b} = \left[\begin{smallmatrix}1&0\\0&0\end{smallmatrix}\right]$. The Jones matrix of the attached quarter waveplate behind the bottom polarizer

with a $45°$ optical axis to the $x$ axis is $\mathbf{J_q} = 1/\sqrt{2}\left[\begin{smallmatrix}1&i\\i&1\end{smallmatrix}\right]$. The second Wollaston prism works as two polarizers when it splits the incident light into two beams deviate to the right ($+x$ axis) and left ($-x$ axis) respectively, and can be represented using two Jones matrices given by $\mathbf{J_r} = \left[\begin{smallmatrix}0&0\\0&1\end{smallmatrix}\right]$ and $\mathbf{J_l} = \left[\begin{smallmatrix}1&0\\0&0\end{smallmatrix}\right]$, respectively. The Jones matrix of the field widened delay plates has a



fairly complex expression, which is described by Langille et al. (2020). Here it is represented by $\mathbf{J_f}$. Thus the output electric fields of the four frames on the CCD detector is given by

$$\mathbf{E_1} = \mathbf{J_r} \cdot \mathbf{J_f} \cdot \mathbf{J_t} \cdot \mathbf{E_t} \tag{6}$$

$$\mathbf{E_2} = \mathbf{J_l} \cdot \mathbf{J_f} \cdot \mathbf{J_t} \cdot \mathbf{E_t} \tag{7}$$

$$\mathbf{E_3} = \mathbf{J_l} \cdot \mathbf{J_f} \cdot \mathbf{J_q} \cdot \mathbf{J_b} \cdot \mathbf{E_b} \tag{8}$$

$$\mathbf{E_4} = \mathbf{J_r} \cdot \mathbf{J_f} \cdot \mathbf{J_q} \cdot \mathbf{J_b} \cdot \mathbf{E_b} \tag{9}$$

In the case of perfect alignment, the Jones matrix $\mathbf{J_f}$ varies little with azimuthal angle $\phi$ of the incident light through the field widened delay plates. To simplify the matrix $\mathbf{J_f}$ and obtain characteristic expressions for the intensity in each quadrant, we assume the incident plane lies along $x$ axis and that the incident angles are small. Substituting into Eq. (6) to (9) and then calculating the average intensity for each beam, the samples in each frame are given by:

$$I_1 = \frac{1}{4} \left[ 1 + \cos\left(\Phi_1\right) \right] \tag{10}$$

$$I_2 = \frac{1}{4} \left[ 1 + \cos\left(\Phi_2 + \pi\right) \right] \tag{11}$$

$$I_3 = \frac{1}{4} \left[ 1 + \cos\left(\Phi_3 + \frac{\pi}{2}\right) \right] \tag{12}$$

$$I_4 = \frac{1}{4} \left[ 1 + \cos\left(\Phi_4 + \frac{3\pi}{2}\right) \right] \tag{13}$$

where $\Phi_i$ $(i = 1, 2, 3, 4)$ is the background phase in each quadrant determined from Eq. 5. With different incident angle $\theta$ and azimuth $\phi$, $\Phi_i$ has different value in the four frames and varies across the field of view resulting in a variation of the phase steps across the scene. In practice, the transmission and the instrument visibility will also vary across the image in each quadrant. If we assume the relative intensities and the instrument visibilities in various measurements are fixed, for a single point, the intensity $I_i^j$ of the $i$th step (frame) in the $j$th measurement can be modeled as

$$I_i^j = K_i I_0^j \left[ 1 + U_i V^j \cos\left(\Phi^j + \varphi_i\right) \right] \tag{14}$$



**Table 4.** Primary science requirements for the BIDWIN prototype instrument.

| Parameter | Requirement |
|---|---|
| Target emission | $O^1D$ at 630 nm, 100 Rayleigh |
| Uncertainty | $< \pm 5$ m/s |

where $I_0^j$ is the mean intensity, $V^j$ is the line visibility. A slight phase shift associated with a Doppler shift in a moving

source can be extracted from these samples using general fringe analysis algorithms if the relative intensities $K_i$, instrument visibilities $U_i$ and phase steps $\varphi_i$ $(i = 1, 2, 3, 4)$ are known. Therefore, these parameters must be carefully calibrated. Another important consideration is thermal drift due to the dependence of the birefringence on temperature, as well as, variations in length to thermal expansion and contraction. In practice, calibration measurements must be performed frequently enough to track the thermal drift.

**4   Prototype instrument design**

**4.1   Overview**

The prototype BIDWIN instrument has been developed for lab performance evaluation and ground-based field testing. Due to the chromatic dispersion of the waveplates in the system, the instrument can only be optimized for operation at a single wavelength. Another important constraint on the design is the acceptable range of possible effective path differences for a

compact field widened birefringent delay plate. This is determined by two factors. First, the magnitude of the birefringence and the availability of large format high quality crystals limits the fixed path difference to the range, 0 to 2 cm. Second, the overall sensitivity of the device to the measurement of Doppler winds is optimized when the normalized wind uncertainty reaches a minimum as shown in Fig. 1 and Fig. 2. From these Figures, we see that only the $O^1S$ and $O^1D$ emissions have normalized uncertainty minima below $D = 5$ cm. Both emissions have similar emission rates when observed from the ground; however,

the $O^1S$ emission at 557 nm corresponds to a layer from 95 km – 110 km, whereas the $O^1D$ emission corresponds to a layer between 150 km to 300 km. In addition, the dynamical motions in the upper layer result in LOS Doppler winds that are on the order of a few 100 m/s, whereas, typical motions in the lower layer result in LOS Doppler winds on the order of 10 m/s.

For the prototype version of the instrument, the design is optimized to target the $O^1D$ emission at 630 nm. At the associated heights, horizontal and vertical motions with velocities on the order of several hundred m/s and 50 m/s respectively and

perturbations on the order of 10 m/s occur. Measurements with uncertainties better than $\pm 5$ m/s are needed to advance our scientific understanding of neutral motions at these heights. Representative emission rates observed at the ground from the $O^1D$ emission varies through the course of a day and also has a seasonal dependence. However, typical emission rates are expected to be near 100 Rayleigh. Optimization of the instrument for 630 nm also allows for lab testing and characterization work to be performed using a stabilized He-Ne laser emitting 632.8 nm. This is ideal since it provides a high signal to noise



ratio, stabilized source with a fixed polarization. The overall requirements that are used to optimize the prototype design are listed in Table 4.

## 4.2 Interferometer design

The primary practical considerations driving the interferometer design are the effective path difference, the SNR and the resulting sensitivity for the measurement of Doppler winds. Several additional criteria were also used to constrain the design

the field widened birefringent delay plate. These include the cost, availability and workability of large format high quality birefringent crystals, the magnitude of birefringence and the thermal stability of the design. LiNBO$_3$, YVO$_4$ and CaCO$_3$ were investigated for the design of the field widened element. YVO$_4$ and CaCO$_3$ achieve a larger path difference compared to Lithium Niobate due to their larger birefringence; however, large aperture YVO$_4$ crystals are difficult to obtain, and CaCO$_3$ is extremely difficult to work with in practice due to its softness. On the other hand, large format LiNBO$_3$ crystals are readily

available allowing for a large throughput device to be constructed. Another important consideration is the thermal stability. All of the crystals examined have strong thermal sensitivities on the order several fringes per degree Celsius change in temperature, resulting in associated wind variation of $\sim 10^3$ m/s to $\sim 10^5$ m/s. Therefore, careful consideration is given to tracking the thermal drift of the instrument during lab testing.Thermally compensated designs are possible (Hale and Day, 1988) and are under consideration; however, this aspect is not considered for the design presented in this paper.

The specifications for the BIDWIN prototype interferometer are shown in Table 5. The interferometer is constructed from two equal length slabs of LiNBO$_3$ that have dimensions 4 cm × 5 cm × 5 cm. The manufacturer guaranteed the optical quality (surface flatness, scratch dig ect.) across a clear aperture of 30 mm centered on the optical axis of the slabs. The true zero order half-wave plate utilized in the system is a 25.4 mm clear aperture element constructed from a birefringent polymer cemented between two slabs of BK7 manufactured by Meadowlark optics. The thickness of the half-wave plate element,

including the mounting, is 1.05 cm. It is optimized for operation at 632.8 nm and has a thermal dependence of the retardance of $\sim 0.15$nm/°C and an angular sensitivity of $< 1\% \pm 5°$. The effective path difference is D = 0.67 cm.

The maximum throughput that can be obtained with the device is fixed by the geometry of a solid block of glass as shown in Fig. 4. The figure shows an incident ray passing through a simple rectangular slab of birefringent material. To achieve as large a signal as possible, the instrument should be designed so the product $A\Omega$ is as large as possible, within whatever restrictions

exist. The usual way to send light through an interferometer is to place a telescope in front that defines the field of view and passes a well-defined collimated beam through the interferometer with an image of the entrance aperture located at the center of the device, which has been discussed in section 2.2.

Assuming a square field of view and a circular entrance aperture, the product $A\Omega$, called the étendue, is given by

$$A\Omega = (\pi r^2) \left[ \tan^{-1} \left( \frac{R-r}{t/2n} \right)^2 \right] \tag{15}$$

where $R$ and $r$ are the clear extent of the slab and the radius of the image of the aperture respectively. The maximum $A\Omega$ is achieved when the image of the entrance aperture midway through the interferometer is half the diameter of the available area at the ends of the interferometer. Taking into account the full length of the assembly (t = 9.05 cm), substituting $R = 15$ mm into



**Table 5.** Specifications of the Lithium Niobate slabs used to construct the field-widened birefringent delay plate.

| Parameter | Specification |
|---|---|
| Individual LiNBO$_3$ slab dimensions (lxwxh) | 4 cm × 5 cm × 5 cm |
| Optical axis angle | 45° |
| Clear aperture | 30mm |
| Coating | MgF$_2$ |
| Refractive index (632 nm) | $n_e = 2.2028, n_o = 2.2866$ |
| Thermal expansion coefficient | $\alpha = 15 \times 10^{-6}/°C$ |
| Birefringence ($B$) | $n_e - n_o \approx -0.0838$ |
| d$B$/d$T$ | $3.96 \times 10^{-5}/°C$ |
| Surface flatness | $\lambda/8$ (over 25.4mm) |
| Scratch/Dig | 40/20 |
| Parallelism | $30''$ |
| Design wavelength | 632.8 nm |
| Half-wave plate | Zero-order birefringent polymer |
| Half-wave assembly thickness | 1.05 cm thick |
| Effective optical path difference | $D = 0.67$ cm (at 632 nm) |
| Maximum throughput | $A\Omega = 0.215$ cm$^2$sr |
| Instrument visibility | $\sim 0.9$ |

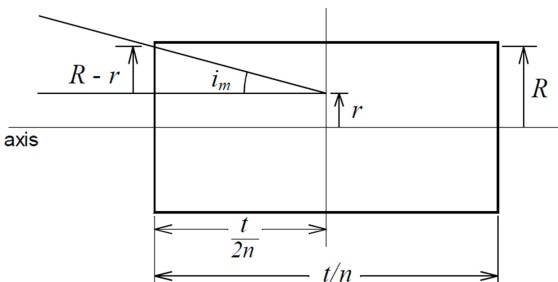

**Figure 4.** The rectangle represents the interferometer, of thickness $t$. A ray enters near the edge at incident angle $i_m$ and reaches midway at distance $r$ above the axis. $R$ is the available radius at both ends of the interferometer. The thickness is shown as air equivalent, $t/n$, where $n$ is the refractive index.





this equation and assuming $n \sim 2.2$ gives a maximum possible throughput of 0.216 cm$^2$sr. This corresponds to a maximum off axis angle through the clear aperture of roughly 20.03 degrees. To mitigate the potential for clipping we limit this to 20

degrees resulting in a solid angle of the square field of view of 0.122 steradians and a corresponding throughput of $A\Omega = 0.215$ cm$^2$sr for the field widened element. Note that if one obtained crystals with high quality across the full 5 cm aperture then the maximum throughput can be significantly increased. In practice, this can be achieved; however, the cost of manufacturing the slabs also increases.

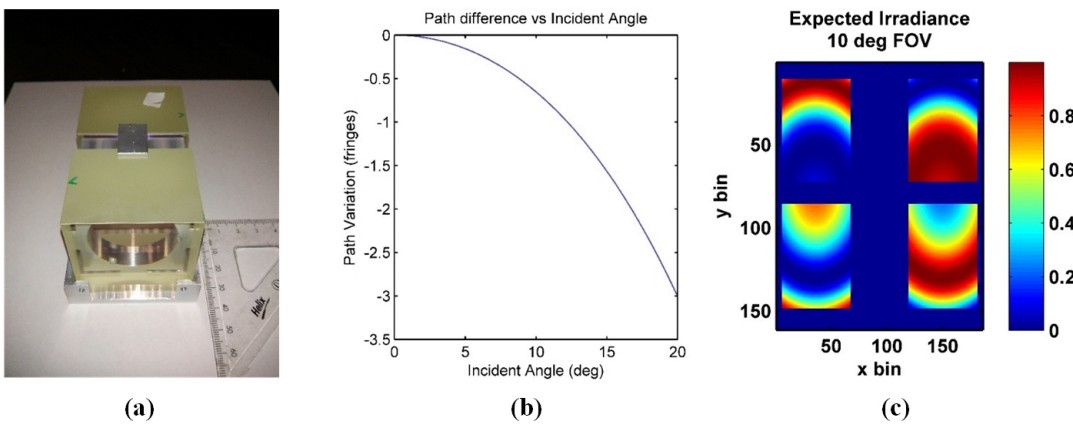

**Figure 5.** The assembled LiNBO$_3$ field widened delay plate (a), optical path variation with incident angle (b), and (c) is a simulated four-quadrant image assuming the configuration discussed in Section 3.

A picture of the assembled prototype is shown in Fig. 5(a). The fully assembled element is roughly 10 cm in length including

the mounting. The optical path variation as a function of incident angle for the device is shown in Fig. 5(b). The extent of the field widening is clear – less than 3.5 fringes enter the field of view with incident angles of 20 degrees. As an example, the simulated interference image produced using this device as the delay plate in the in the BIDWIN optical system assuming a range of off axis angles of 10 degrees is shown in Fig. 5(c). In this simulation, the detector is taken to have 250 x 250 bins and ideal polarization selection by the Wollaston prisms is assumed. The phase variation across the image of the scene and

the associated quadrature between the samples in the image is shown in Fig. 6(a)-(c). The upper left panel shows the phase across the top quadrant and the upper right panel shows the phase variation across the bottom quadrant. The associated phase quadrature is shown in the bottom left panel. It is clear that there are several "strips" across the image that have zero phase quadrature. The expected wind uncertainty is shown in Fig. 6(e) as a function of the SNR. Achieving a wind precision of less than 5 m/s requires an SNR > 700.

The expected wind uncertainty across a $61 \times 61$ bin image of the scene is examined in Fig. 6(d). This was obtained by performing Monte Carlo simulations assuming realistic signal levels with Poisson noise added to the measurements (SNR = 700). The four-point algorithm cannot be applied to the "strips" with zero quadrature which results in enhanced wind errors within these regions. This effect is enhanced as the field of view is increased; therefore, care must be taken to properly identify





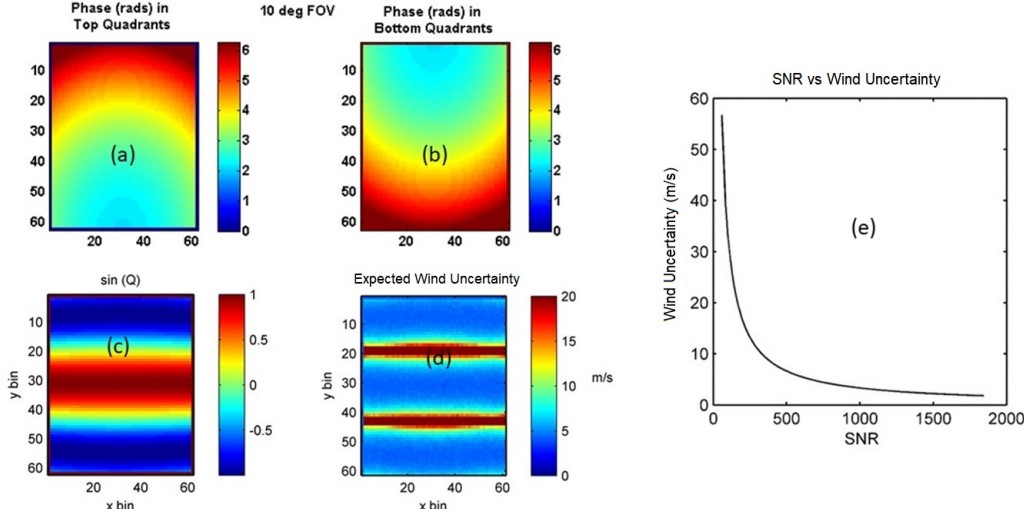

**Figure 6.** The expected phase variation in the top (a) and bottom (b) quadrants of the BIDWIN instrument and the phase quadrature between samples (c) assuming a 10-degree field of view. The expected wind uncertainty (d) and (e) assuming the instrument characteristics listed in Table 5 and the signal characteristics listed in Table 4 and Table 2.

and characterize these regions during data processing.However, the presence of these zero quadrature regions also provides an
opportunity to simultaneously sample the intensity at several positions in the scene. The potential application of this feature for a potential limb viewing satellite version of the instrument is presented in the Discussion section.

## 4.3 The imaging system

The side view ray trace through the BIDWIN optical system is shown in Fig.7. As discussed in the previous section, the throughput of an ideal system is limited by the maximum throughput that can be provided by the delay plate. However, in the
case of the breadboard instrument it was the size and availability of large format Wollaston prisms that limited the maximum throughput. The prisms utilized in the breadboard system are constructed from two $YVO_4$ wedges. The wedges are designed to provide a split angle of of $\sim 9.29°$ (at 632.8 nm) between the two orthoganl polarizations exiting the Wollaston. The objective lens is designed to accept an input field of view of 4 degrees square and the 10 mm diameter pupil of the prism located directly in front of the objective lens defines the entrance aperture of the optical system.

The objective lens was optimized such that the output beam is approximately telecentric which minimizes the incident angle at the waveplate/split field polarizer location. This is done to reduce the impact of the incomplete polarization selection of off axis rays at the polarizer and slight shifts in the retardance of the quarter waveplate due to the angular dependence of the retardance. The collimating lens is designed to pass a collimated image of the field stop through the field-widened birefringent delay plate. The collimating lens also forms an image of the entrance aperture midway between the birefringent delay plate.





The imaging lens is optimized to correct for aberrations and focus the collimated beams onto the detector. All of the lenses are spherical and are constructed from BK7. The lenses have been anti-reflection coated for visible wavelengths.

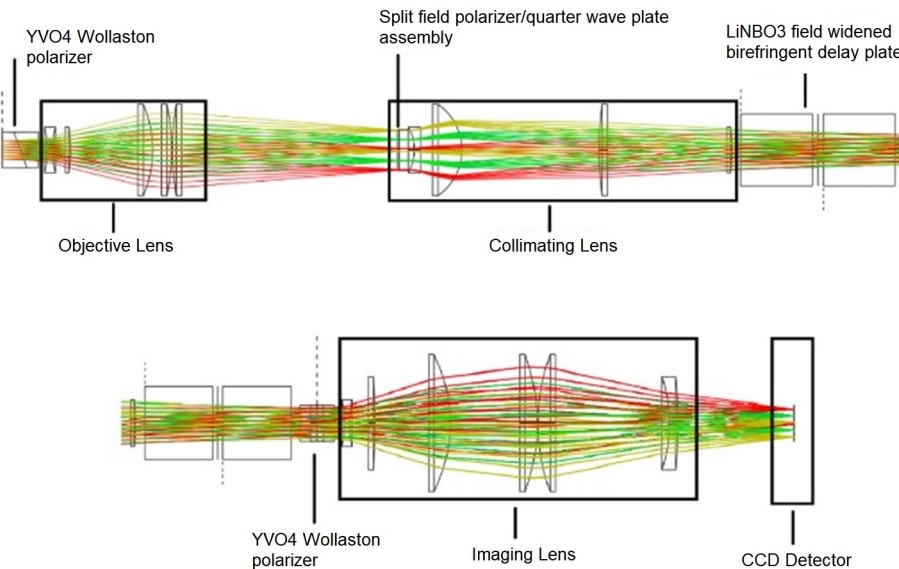

**Figure 7.** The BIDWIN optical design (Side view ray trace).

   A comparison between the interference fringes simulated using Zemax optical design software (upper panels) and the observed fringes (lower panels) in the lab is shown in Fig. 8. Three examples are shown. The first example ((a) and (d)) shows the case of ideal alignment between the components. We observe less than one fringe in the field of view and the images are phase

stepped in 90-degree increments. Because of the breadth of the fringe, its form in the lower panels is more difficult to see. The shape of the field stop that located near the split field polarizer is observed in the lower panel and the edges of the split field polarizer can also be seen. In the second example, the back crystal has been rotated by 10 degrees. This misalignment introduces high contrast hyperbolic fringes as well as a set of low amplitude parasitic fringes. The parasitic fringes are the result of misalignment relative to the halfwave plate and are removed in the third example by rotating the halfwave plate by 5 degrees.

All three cases agree extremely well with the simulated fringes and serve to demonstrate the sensitivity to misalignment as well as the overall imaging quality of the optical system.





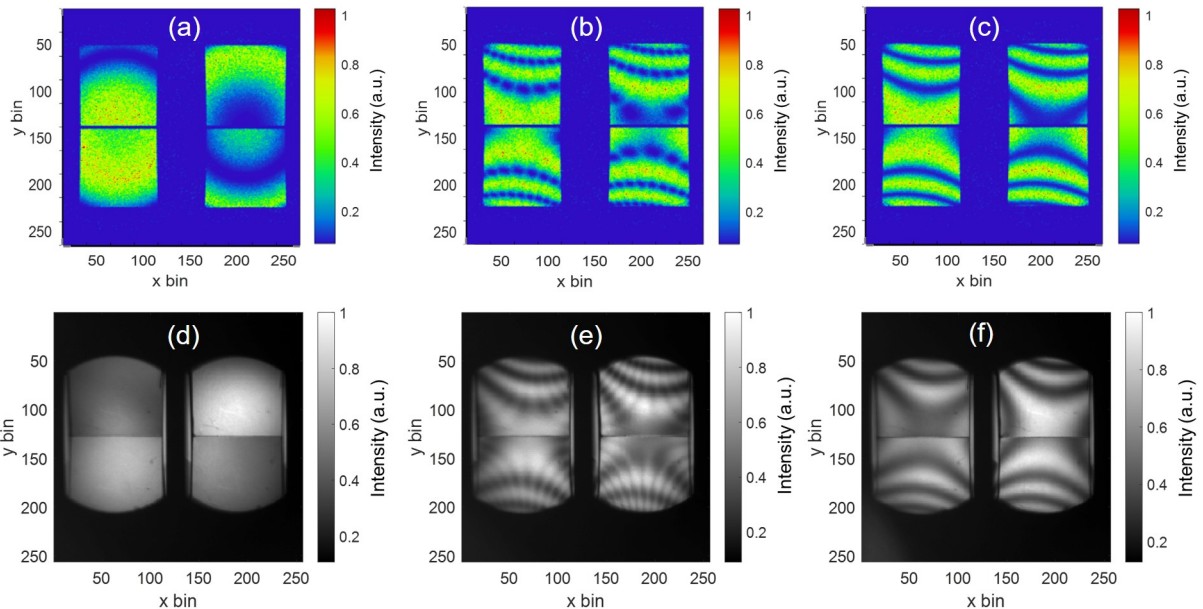

**Figure 8.** Comparison between the simulated (upper panels) and observed (lower panels) interference fringes for the case of perfect alignment (a, d), a 10-degree misalignment between the Lithium Niobate slabs (b,e) and the same as in (b,e) with the half-wave plate rotated by 5 degrees to eliminate the low amplitude parasitic fringes.

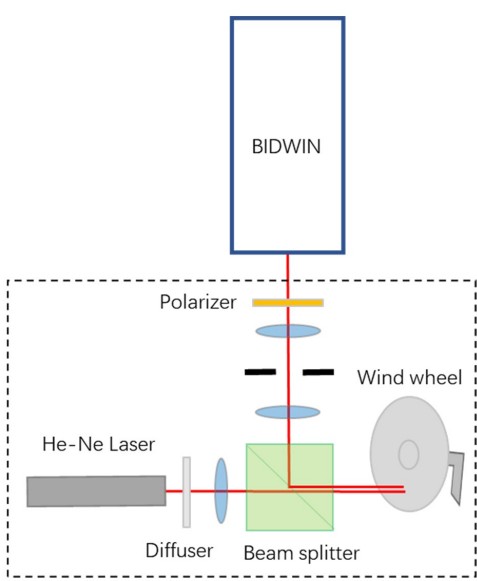

**Figure 9.** Schematic of the system used to produce a predicable gradient in the LOS wind field within the field of view of the instrument in the lab.



## 5  Lab Performance

### 5.1  Setup

The BIDWIN prototype instrument was assembled at the Atmospheric and Space Physics Lab at the University of New Brunswick. The basic imaging configuration for this comparison is shown in Fig. 3. The system used to produce a predictable gradient in the the LOS wind field within the field of view of the instrument is shown in Fig. 9. Light incident from a He-Ne laser is diffused and passed through a beam splitter to illuminate a retro-reflective disk that was oriented at an angle of $45°$ to the optical axis. The disk was attached to a chopper and controller system with which the rotation rate of the wheel was accurately controlled. Light retro reflected from the disk is reflected by the beam splitter and collimated before collected by the

BIDWIN entrance optics. Light emitted from the He-Ne laser is only partially polarized with a narrow range of polarization's near some specific angle of polarization close to the direction of the y-axis.Therefore, a polarizer oriented at $45°$ to $x$ axis was placed in front of the system to ensure the two beams split by the first Wollaston prism are equal in intensity. This polarizer is not required when observing the unpolarized airglow. The first Wollaston prism is defined as the entrance aperture, which has a 10mm clear aperture. An Apogee U47 CCD with a resolution of $256 \times 256$ on $2 \times 2$ binning is used for imaging. The imaging

optics part of the lab prototype was configured slightly different from Fig. 3. A folding mirror and lens are used to reimage the primary interference fringe image to ensure the four frames match the size of the CCD that was available for lab testing. In addition, the field of view of the optical system, set by an intermediate stop in the wind wheel system, is a 4 degrees circle instead of a square field of view.

### 5.2  Characterization and calibration

The calibration of the fringe parameters as discussed in Section 2 can be achieved by scanning the wavelength of a suitable isolated spectral emission line, scanning fringes through temperature variations (i.e. using the thermal dependence of the glass properties to change the path) or by rotating the field widened delay plates (Gault et al., 2001) and sampling the interference pattern at each step. In our experiment, scanning the wavelength of the frequency stabilized 632.8nm He-Ne laser was not possible, so we utilized the strong thermal dependence of the Lithium Niobate slabs to scan the optical path.

Because of the thermal sensitivity of the field widened delay plates, a scan in the optical path of almost two fringes can be performed by letting the system respond to the the variation of the lab temperature over roughly 30 minutes as shown in Fig. 10. It is obvious that the frequency of the cosine curve is changing in the scan, suggesting that the variation of the lab temperature is not linear. Thus the curve fitting method does not work here for the calibration of the fringe parameters. The LMS algorithm initially developed by Ward (1988) and refined by Kristoffersen (2019); Kristoffersen et al. (2021) to

simultaneously determine both the phase steps associated with sampling a fringe and the fringe parameters of the emission can be applied to the BIDWIN situation. In this case, the instrument fringe parameters from every bin in each quadrant must be characterized before standard fringe analysis techniques can be used. This characterization includes determining the relative phases of the four bins viewing the same segment of the scene, the $\varphi_i$, the instrument visibility, $U_i$ and the relative responsivity, $K_i$. An elegant way of visualizing this process is through Lissajous mapping (Yan et al., 2021). The measured intensities of





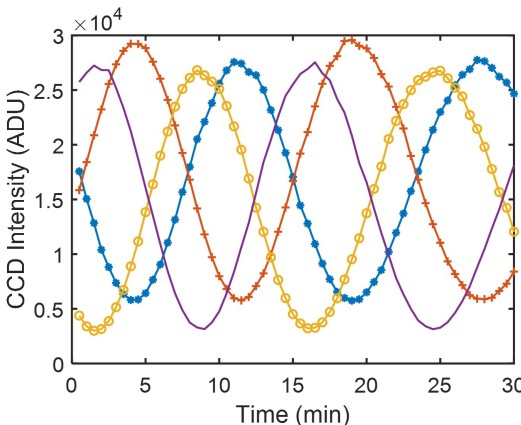

**Figure 10.** The intensities of the centre points measured by thermally scanning of BIDWIN.

two quadrants during a thermal scan of BIDWIN are used to fill the $2\pi$ phase space of a fringe and in a least-mean square sense determine the ellipse associated with this measurement set. Using the parameters of the ellipse, $K_i$, $U_i$ and $\varphi_i$ can then be calculated. Applying this algorithm to all of the points in the field of view, we can acquire two-dimensional images of the calibrated fringe parameters: $K_i$, $U_i$ and $\varphi_i$. The fringe parameters obtained using this approach are shown in Fig. 11.

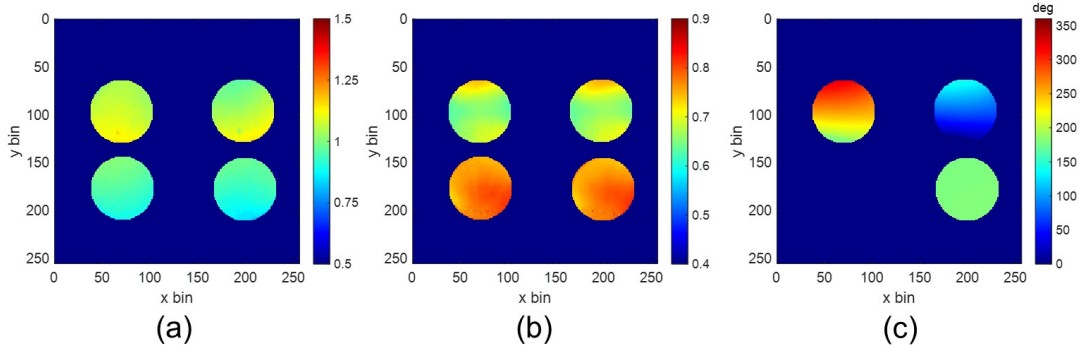

**Figure 11.** The calibrated parameters over the field of view of BIDWIN. (a) relative intensities $K_i$; (b) instrument visibilities $U_i$; (c) phases $\varphi_i$ relative to the first quadrant.

Before applying these calibration parameters to the observations, the phase order of four quadrants must also be determined.
Ideally, the steps should be in $90°$ increments. According to the Jones matrix model, the phase steps come from the four combinations of polarizers and waveplate. So we can match those combinations with the four quadrants on the detector by observing the intensities of the four quadrants after removing the field widened delay plates. After the phase order determination, it was found that the quadrant of the first phase step is the left bottom one because of the added folding mirror and lens in the imaging





system. Flat field images were also obtained using this configuration by placing a polarizer oriented at $45°$ at the position of
the field widened delay plates.

Figure 11 shows the results of the $K_i$, $U_i$ and $\varphi_i$ calibration. In Fig. 11(a), the relative intensities of the top quadrants are
slightly higher than the bottom quadrants. Besides the misalignment of the optical train, a possible source of this variation
is the angular error of the polarizer in front of the system, which should be oriented at exactly $45°$ to ensure the two split
light beams from the first Wollaston prism are equal. The instrument visibilities are mainly affected by features of the Lithium
Niobate plates, such as the surface flatness and uniformity. As is shown in Fig. 11(b), unlike the scanning mirror but similar to
the segmented mirror Michelson interferometers, the visibilities of the four quadrant samples exhibit strong differences. It is
obvious that the top and bottom light beams propagate through different parts of the crystal and result in different visibilities.
In our experiment, we found that the visibilities of the four quadrants varied upon translation of the Lithium Niobate plates
perpendicular to normal incidence. This suggests some spatial path variations that are larger (or smaller) for different regions
of the crystal surface.

The calibrated phases shown in Fig. 11(c) are the phase differences of the four quadrants relative to the first one (lower
left quadrant which necessarily will have a relative phase of zero). The phase differences between the two horizontally spilt
quadrants of top and bottom should be exactly $180°$. However, the phase differences between the top and bottom quadrants
vary across the field of view. This is because the upper and lower beams separated by the first Wollaston prism form identical
images of the scene within the top and bottom quadrants of the split field polarizer. The field from the top and bottom quadrants
is then passed as collimated light through the field widened delay plates - mapping direction through the plate to position on
the field stop. Therefore the phase in the upper quadrant is mapped to position (and associated direction through the plate) in
the top portion and the phase in the lower quadrant is mapped to position in the bottom quadrant. Therefore, the phase steps
between the upper and lower neighbouring quadrants have a vertical distribution in the field of view and the $\sin(|\varphi_1 - \varphi_2|)$ is
shown in Fig. 12(a).

The ideal phase step for the four-point algorithm generally used for fringe sampling is $90°$ (Shepherd, 2002), relative phase
steps deviate increasingly from this ideal with increasing incident angle (field of view). This results in larger uncertainties in
the retrieved Doppler shifts as discussed in Section 4 (see (Kristoffersen et al., 2021) for a detailed discussion of the effect
variations in phase steps have on fringe parameter determinations). The simulated uncertainty distribution for the current
configuration was simulated using the calibrated $K_i$, $U_i$ and $\varphi_i$ using a Monte Carlo method. In the simulation, shot noise was
added to the signal with a SNR of 1000. The result is shown in Fig. 12(b). Observe that bins close to the top and bottom edge
have greater uncertainties. Therefore, the region of the field of view that can be used to achieve a wind precision less than 5m/s
is only slightly restricted here.

### 5.3 Lab wind measurements

Two sets of wind measurements were conducted in the lab to examine the performance of the BIDWIN system. The first
experiment is a single point wind measurement. This involved observing a specific point on the wind wheel and rotating the
wind wheel at different rates so that a series of velocities were observed. This experiment tested the Doppler shift measuring

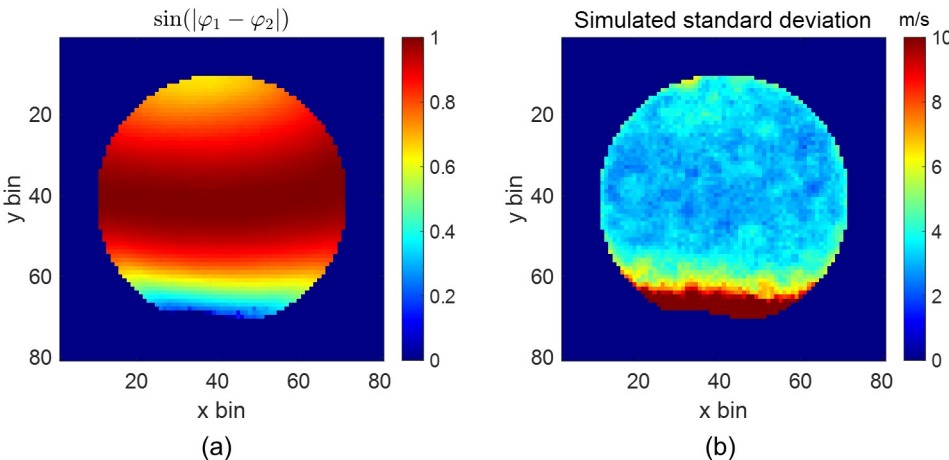

**Figure 12.** The sine value of the phase steps between $\varphi_1$ and $\varphi_2$ and the simulated standard deviation using calibrated parameters. (a) the $\sin(|\varphi_1 - \varphi_2|)$ distribution in the field of view; (b) simulated standard deviation in retrieved wind.

capacities of BIDWIN without including the imaging capability. The second one is a two-dimensional wind measurement which is undertaken by imaging an area of the wind wheel when it is rotating at a certain specified frequency.

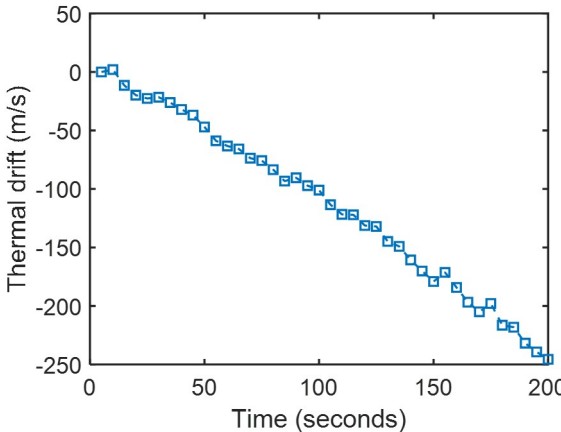

**Figure 13.** Thermal drift of BIDWIN over 200 seconds interval using He-Ne laser as the source.

To make a wind measurement, phase measurements of the rotating wheel and stationary wheel are required. The phase measurements with the stationary provide the zero wind background phase. Wind measurements are determined from the phase difference between this phase and the phase of the rotating wheel. Without a good thermal enclosure, frequent measurement of this background phase is essential for these experiments because the system is highly sensitive to temperature (the thermal drift corresponds to roughly 1.25 m/s per second in the experiment, see Fig. 13). Unless carefully taken into account, this drift can





dominate the wind determinations. In our experiment, we removed the background phase by linear fitting. The measurement
sequence for Doppler wind measurement is similar to the measurement approach for MIADI, in which the zero wind image
was taken after each wind measurement (Langille et al., 2013b). The time of each wind measurement was recorded and the
background phase of that moment could be interpolated.

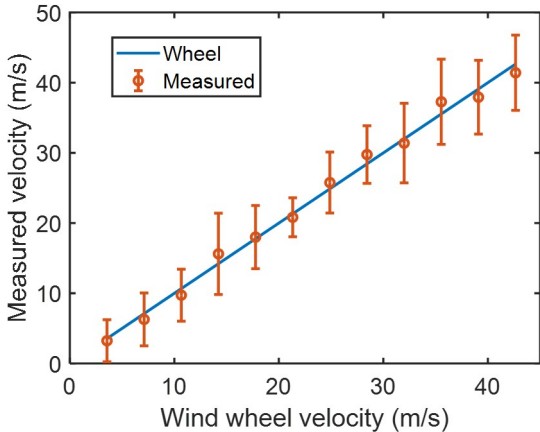

**Figure 14.** Measured wind velocity for single point measurements is plotted versus the wind wheel velocity.

For the single point experiment, the distance from the wheel center to the center point of the narrow field of view was 4 cm.
Phase determinations were made using the 30 by 30 pixel region illuminated on the CCD, resulting in an irradiance SNR of ∼
1000. The frequency of wheel rotation was adjusted incrementally from 10 Hz to 120 Hz in steps of 10 Hz to provide a range
of line-of-sight speeds to evaluate the wind determination. The wind velocity measured using BIDWIN is plotted versus the
wind wheel velocity determined from the rotation rate in Fig. 14. The straight blue line is the expected velocity. The red circles
are the average measured velocity of nine measurements and the error bars are the standard deviations. The average standard
deviation of the twelve points is 4.53 m/s, which confirms that upper atmospheric Doppler wind measurements with precision
of 5 m/s are feasible using this technique.

The imaging capability was examined by illuminating a 2 cm diameter circular area centered 4 cm from the center of the
retro-reflecting disk. The position of each pixel relative to the center of the wind wheel was determined by imaging a grid scale
printed on a circle transparent plastic sheet which had a same size as the disk. Six measurement sets were taken using a rotation
frequency of the disk of 105 Hz. The exposure time was adjusted to get a SNR of 1000 by averaging neighbouring pixels.
Because all bins in the wind field are measured simultaneously, there is no thermal drift across the field of view; therefore, the
thermal drift calibration for one point can be applied to the whole field while removing the background phase. The expected
wind field is shown in Fig. 15(a) and the average of the six BIDWIN Doppler wind field measurements is shown in Fig. 15(b).
The velocity gradients of the two wind images are consistent in shape and in magnitude. Some spatial variability is observed
that does not track the gradient; however it is possible that this is associated with contamination from light scattered from the
disk that is not perfectly retro-reflected due to spatial variations across the disk.





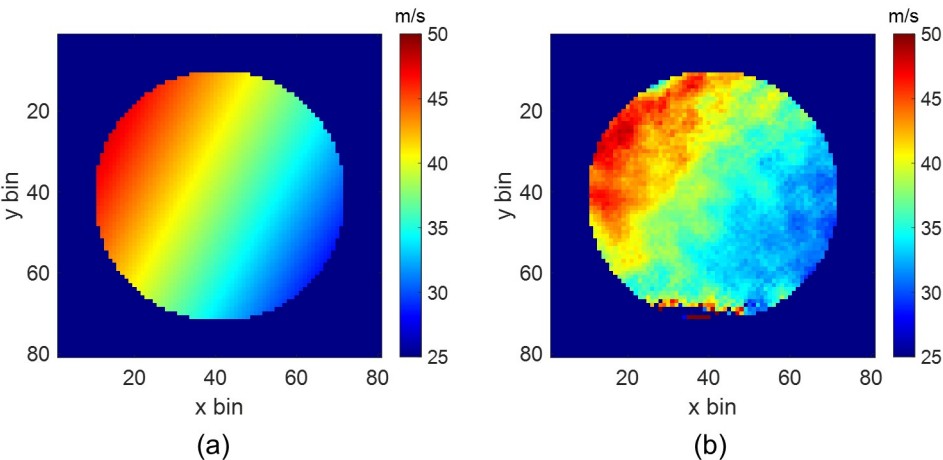

**Figure 15.** The expected and measured two-dimensional wind field across the wind wheel. (a) expected wind field; (b) measured wind field.

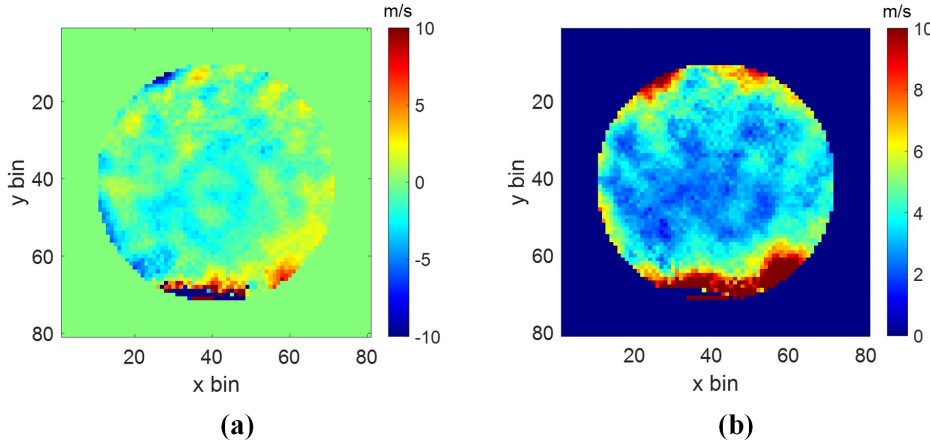

**Figure 16.** The difference between the expected and measured wind field is shown in (a) and the velocity standard deviation of the measured wind field across the field of view is shown in (b).





The difference between the expected and measured wind field is shown in Fig. 16(a). Across most of the field of view, the difference between the two fields is less than 2 m/s. As anticipated, the errors in the top and bottom edges are much larger because the phase steps move away from quadrature. The velocity standard deviation of the measured wind field is shown in

Fig. 16(b). It is consistent with the simulated standard deviation image shown in Fig. 12(b). The usable field of view is only slightly restricted by regions near the top and bottom edges where the wind error rapidly increases as the phase steps deviate significantly from quadrature. The system achieves a precision of better than 5m/s across a large portion of the field of view.

This standard deviation is slightly higher than that predicted using the Monte Carlo simulations. There are some possible error sources such as the impact of uncertainties in the calibrated instrument parameters, the stability of the laser, the residual

errors of the thermal drift calibration and the presence of scattered laser light from the beamsplitter. The analysis of these error sources was not performed for this study and will be undertaken in the future.

## 6 Discussion

The optical configuration of the breadboard prototype described in this paper has been used for lab testing, characterization and performance evaluation. Single point wind measurements and two dimensional wind measurements have been performed

in the lab in order to examine the feasibility of the technique for the measurement of upper atmospheric winds. Using these measurements it was shown that wind precisions of < 5 m/s are achievable with the interferometer. This work has also facilitated the identification of the primary practical issues that must be carefully considered in order to implement this instrument in the field. These include the following:

1) The sensitivity of the instrument to the field of view, misalignment of and mismatches in birefringent components.

2) The loss of quadrature across regions of the image which results in increased wind uncertainties within these regions.

3) Thermal sensitivity of the field widened birefringent interferometer.

4) Availability of high quality large format Wollaston prisms and uniaxial birefringent slabs.

The impact of misalignments and mismatches was briefly discussed in Section 4. As shown in Fig. 8, as the field of view is increased, horizontal strips across the image are introduced that exhibit zero quadrature. Rotational misalignment between

the Lithium Niobate slabs introduces hyperbolic fringes which changes the shape and increases the number and size of these zero quadrature regions. Additionally, misalignment or manufacturing errors in the halfwave plate introduce parasitic fringes due to unequal coupling between the e and o waves in the interferometer. The overall impact of these sensitivities and the loss of quadrature between frames on the wind measurements must be investigated. This includes evaluating their effect on the accuracy of the calibrated fringe parameters, which also affect the precision of the wind measurements.

In the case where the loss of quadrature within certain bands is present, we envision arranging a limb viewing satellite instrument such these bands are projected perpendicular to the horizon. In this case, there is only a loss of wind information over a small range horizontal bins at each tangent altitude. While these regions cannot provide wind information, they will provide simultaneous intensity profiles along the vertical dimension. A Jones matrix framework has been developed that allows these sensitivities to be examined by accurately simulating the interference fringes observed with the instrument (Langille et al.,





2020). This framework was used in the design presented in Section 4 and provides a pragmatic and efficient means to evaluate and implement further refinements to the design and measurement approach.

The current BIDWIN configuration exhibits a strong thermal dependence that is dominated by the change in birefringence with temperature. This issue was managed for the lab measurements by using short integration times, sampling the non-rotating wheel between measurements and interpolating the thermal drift between measurements. This effect would need to

be carefully managed in the case of a practical field instrument where longer integration times are required, by implementing thermal compensation and active thermal control. It may also be possible to partially thermally compensate the field widened birefringent interferometer by combining appropriately selected and oriented slabs so that the temperature dependence of the two composite slabs are oppositely signed (Hale and Day, 1988). The design of a thermally compensated field widened birefringent interferometer requires careful and rigorous consideration that is outside the scope of the current work.

The final consideration relates to the availability of large format high quality Wollaston prisms and Lithium Niobate slabs. For the prototype design, the size of the Wollaston prisms rather than the field widened birefrigent delay plate limited the throughput, and as a result, much smaller than the maximum that is possible given the physical size of the delay plate. To realize the full capability of this design, a practical field instrument will need to utilize this larger throughput by acquiring custom large format prisms. Given the availability of large format crystals, a realistic field widened birefringent interferometer

can be constructed with effective path difference in the 0 to 2cm range with diameters on the order of 100 mm. Therefore instruments capable of achieving a throughput on the order of   1 cm$^2$sr are feasible. Comparison of the effectiveness of a field widened birefringent interferometer relative to a field widened Michelson interferometer for the measurement of Doppler winds can be undertaken with respect to the primary instrument design parameters: $A$, $\Omega$ and $D$.

For this comparison we assume that both instruments are observing the same emission lines using the same integration time

and that the wind precision is dominated by photon noise. By combining Eq. (3) and Eq. (4) the wind uncertainty as a function of the source characteristics and the instrumental parameters is:

$$\sigma_w = \frac{c\lambda}{UVD\sqrt{\frac{10^6}{4\pi}I_0 t\eta A\Omega\tau_c}} \tag{16}$$

The comparison is further simplified by assuming the instrumental parameters - $t$, $I_0$, $\eta$ and $U$ are the same. Since the visibility $V$ is a function of $D$ it must also be included in the comparison. The instruments' relative wind-measuring precision,

$E$, evaluated with respect to throughput, path difference and the line visibility $V$, is:

$$E = \frac{\lambda}{VD\sqrt{A\Omega\tau_c}} \tag{17}$$

In evaluating $A\Omega$, the size of one field of view is used with the corresponding collecting area. Therefore, in the cases where four copies of the image are formed, such as for the instrument discussed in this paper, it is at the expense of the intensity level in each of the four copies. We account for this effect here by multiplying the transmission coefficient of those instrument

by 0.25. Table 6 lists the values of $E$ for several Michelson interferometers, both built and proposed, and for the birefringent interferometer discussed in this paper, for four representative airglow emissions.





**Table 6.** Relative wind precision ($E$) evaluations for Michelson and birefringent interferometers (BI).

| Instrument | $A\Omega$ (cm$^2$sr) | $D$ (cm) | $\tau_c$ | $E$ ($10^{-5}$ cm$^{-2}$sr$^{-1/2}$) | | | |
|---|---|---|---|---|---|---|---|
| | | | | O$^1$S (200 K) | O$^1$D (1000 K) | OH(6,2) (200 K) | O$_2$$^1\Delta$ (200 K) |
| WaMI | 0.12 | 5.9 | 0.25*0.5 | 10.0 | 23.6 | 13.0 | 18.0 |
| DynAMO | 0.040 | 9 | 0.5 | 7.9 | / | / | 10.5 |
| MIADI | 0.091 | 7.45 | 0.5 | 5.3 | 19.5 | 6.3 | 8.3 |
| WINDII | 0.48 (N) | 4.46 | 0.5 | 3.0 | 5.1 | 4.1 | / |
| | 0.053 (D) | | | 8.9 | 15.3 | 12.3 | / |
| Lab: LiNbO$_3$ (BI) | 0.215 | 0.67 | 0.25*1.0 | 36.0 | 41.1 | 54.3 | 81.4 |
| Field: LiNbO$_3$ (BI) | 0.86 | 1 | 0.25*1.0 | 12.1 | 14.0 | 18.2 | 27.3 |

According to this analysis, the prototype LiNbO$_3$ birefringent interferometer has a wind measurement error that is comparable to but larger than that of several current field widened Michelson interferometers. However, a practical field instrument that is equally matched to the field widened Michelson interferometer can be achieved by utilizing slightly larger and longer crystals in order to increase the path difference to D = 1 cm and increase the usable aperture area by a factor of at least 4. It is possible because the diameter of a LiNbO$_3$ crystal can reach 100mm. In this case, the birefringent interferometer can achieve a throughput of 0.86 cm$^2$sr and is capable of achieving similar wind errors and yet it has the smallest path difference. This high precision is a result of the large throughput provided by the large format LiNbO$_3$ crystals. In order to take advantage of this potential, very small phase shifts must be measured with very high precision, using the high signal-to-noise ratios resulting from its large throughput. In addition, utilizing the large throughput provided by the birefringent interferometer requires that the other optics surrounding the interferometer are designed to accommodate it.

## 7 Conclusions

This paper presented the concept, design and performance testing of a compact static birefringent interferometer called BID-WIN. The overall measurement principle, as well as, the optical system and interferometer configuration was described. The design and implementation of the lab prototype was presented and the instrument parameters were carefully characterized and calibrated. The expected wind precision and the limitation of the field of view has been analysed and the performance of the design was evaluated. The feasibility of measuring upper atmospheric winds with precision of 5m/s using BIDWIN was validated by performing single point wind and two-dimensional wind field observation in the lab. The practical limitations associated with the design of a large throughput BIDWIN instrument capable of field measurements was discussed. Further study is needed to take full advantage of the technique; specifically, the ability to accommodate large aperture optical components and to implement thermal compensation. The overall performance of the prototype demonstrates the feasibility of the technique for the measurement of upper atmospheric winds.



*Author contributions.* Scott and Bell: Inital concept and support of the development of the lab instrument; Gault: Initial design; Sheng and Touhiri: Optical design of the lab instrument; Langille: led the conceptual design and analysis, construction of the instrument and lab
measurements; Yan: Performed the lab measurements and analysis, and wrote the paper together with Langille; Ward: Supervised the overall instrument development, testing, analysis and writing; Zhang: All co-authors contributed to the review and discussion of the paper.

*Competing interests.* The authors declare that they have no conflict of interest.

*Acknowledgements.* This research has been supported by National Science and Engineering Research Council industrial post-graduate scholarship (IPS-2); National Natural Science Foundation of China (4202040400, 41530422) and China Scholarship Council (CSC) (201806280438).
Support from the Canadian Space Agency, the Canadian Foundation of Innovation and the National Science and Engineering Research Council for the maintenance of the Atmospheric and Space Physics Lab at the University of New Brunswick is gratefully acknowledged.





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
