# Peer review of "A compact static birefringent interferometer for the measurement of upper atmospheric winds: concept, design and lab performance"

_Atmospheric Measurement Techniques, 2021_

## Author Response (AR1)

**Response to the Referees**

**Author comment:** Thank you for your comments and suggestions. The comments from Referees are shown in black, our responses and our changes in manuscript are shown highlighted in blue and red below.

**Review # 1**

**Comments:**

The following is a review of the manuscript by Yan et al., with the title: "A compact static birefringent interferometer for the measurement of upper atmospheric winds: concept, design and lab performance." The manuscript presents the instrument concept and the design and optimization of a prototype version of the instrument. Characterization of the lab prototype is also presented in addition to the performance of the instrument using a low velocity, laboratory generated, two-dimensional Doppler shifted signal field.

   This manuscript documents the theoretical and numerical model calculations, as well as laboratory measurements, confirming nearly shot noise limited performance of a carefully built laboratory instrument. The instrument concept is not new, but the specific measurements, including the imaging observations are. My main concern with this manuscript is that, especially since this is not a new concept, which would warrant a publication, it is not made clear what the advantages of this approach are, which presumably motivate this work. The authors say that "The arrangement provides a similar throughput to that of a field widened Michelson interferometer, albeit constructed without moving parts. Consequently, the instrument provides a compact, lightweight and robust alternative." However, there are several other Michelson-type interferometers that have been used, are in use, or are in the literature, which are also constructed without moving parts and are compact, lightweight and robust. In addition, this concept, according to the authors, might, in practice, not be able to be constructed with the optimal path difference for the atmospheric application (i.e. emission line width). Given the above, it is my opinion that this paper (a) should be motivated much more clearly, i.e. state quantitatively what advantages over the state-of-the-art (Michelson, Fabry-Perot, DASH,…) are expected from this approach, and (b) should be using the results of this work to provide evidence that these advantages can be achieved. This would make it a well-rounded and valuable contribution to the literature.

**Author's response:**

You make a valid assessment that the implementation described in the manuscript is an adaptation of a technique that was previously applied to measure high speed (many hundreds of m/s) Doppler winds in plasmas (Hunten et al., 2010). However, the optimization for low speed Doppler wind measurements using airglow emissions is definitely a new (and unpublished) concept. The instrument design described in this paper, especially the birefringent delay plate, has been optimized for the measurements of LOS Doppler winds using isolated

airglow emissions. While the optical path difference of the birefringent interferometer cannot be designed with the optimal optical path difference (at the minima in Figure 1), we can realize a version of the instrument with an optical path difference near the minimum (~1 cm). Moving away from the minimum is compensated by constructing an instrument with a large throughput. The resulting interferometer (a field widened delay plate constructed from Lithium Niobate) has a larger path difference and much larger throughput than the instrument described in [Hunten et al., 2010]. Accordingly, many aspects of the optical design have been optimized to accommodate this throughput. Numerous examples of such optimization for a specific application is found throughout the literature in the development life cycle of state of the art techniques, including various versions the field widened Michelson [Langille et al. ,2013, Kristofferson et al., 2017] and SHS [Harlander, 1991, Englert et al., 2020, Kauffman et al., 2020, Langille et al., 2019].

The primary advantage of this technique over state-of-art field widened Michelson, Fabry-Perot and the DASH instruments is the size, volume, and minimal complexity in the construction of the interferometer component. The delay plate component is roughly 10 cm x 5 cm x 5 cm and can be assembled using tools available in most optical labs. The same cannot be said for the state-of-the-art where assembly and construction requires extreme skill and has only be mastered by a handful of industrial players. In fact, industrial involvement in the development of this instrument was primarily motivated by the simplicity of the construction and the ability to by-pass outsourcing the interferometer construction and assembly. We agree that this aspect can be motivated more clearly in the introduction, and we have included a paragraph accordingly.

Accepting the simplicity of the interferometer construction as the primary motivation and advantage over the state-of-the art, the goal is to then provide evidence that such an instrument has similar capabilities to the state of the art. Here, we suggest that the quantitative comparisons presented in the Discussion between existing field-widened Michelson interferometers and the BIDWIN instrument provides adequate evidence. More broadly, with the paper more clearly motivated in the introduction, we argue that the material presented in the paper provides a valuable contribution to the literature.

**Author's changes in manuscript:**

We have added the content in Section 1 and Section 2.2, please see line 62 to line 68, and line 164 to line 169 in the marked-up version.

**Comments:**

Some specific comments are:
(1) Table 1: please provide references for the projects so the readers can follow-up. Especially for the proposals and studies, which are traditionally hard to find (or leave those out). In addition, there are many other ground based Fabry-Perot wind interferometers, please at least mention that fact.

**Author's response:**

Table 1 has been updated with an additional column that includes the specific references.

**Author's changes in manuscript:**

The references have been added to Table 1, please see line 105 to line 109 in the marked-up version.

**Comments:**

(2) Table 2: the tangent heights for O1D are missing.

**Author's response:**

The tangent height (250 km) has now been included.

**Author's changes in manuscript:**

The tangent height has been added to Table 2, please see line 125 to line 130 in the marked-up version.

**Comments:**

(3) Line 91: the authors state: "The usual way to send light through an interferometer is to place a telescope in front that defines the field of view and passes a well-defined beam into the interference optics, with an image of the entrance aperture half-way through the interferometer." It is not clear to me that the statement about the location of the image position is generally true. In addition, this statement does not contribute to the overall message of the paper, so I recommend omitting it.

**Author's response:**

This line has been omitted in the revised version.

**Author's changes in manuscript:**

The statement has been omitted, please see line 140 to line 143 in the marked-up version.

**Comments:**

(4) The authors state at several places that knowledge of thermal instrument drift is essential. Please specify if there is a plan to measure the thermal drift simultaneously with the atmospheric measurement (and if so, how), or if it has to be done sequentially.

**Author's response:**

We have two options. First, we can, as you suggest, periodically observe a calibration source emitting a spectral line close to that of the target emission. In this case, the requirements on the thermal stability are high. The second option, and the one we plan to implement is to place a dichroic beam splitter in the aft-optics which will produce a copy of the four-quadrant image on a second detector. In this case, light from a calibration source at a slightly different wavelength will illuminate the front optics simultaneously to the scene radiance. This will allow for the thermal drift to be monitored simultaneously to the atmospheric measurement.

**Author's changes in manuscript:**

We have added the discussions to Section 3, Section 4.2, and added the following lines to the Discussion. Please see line 239 to line 247, line 286 to line 287 and line 576 to line 580 in the

marked-up version.

"This effect would need to be carefully managed in the case of a practical field instrument where longer integration times are required, by implementing thermal compensation and active thermal control. In this case, the thermal drift will be tracked (and then corrected) by observing a calibration source. This could be done periodically between scene measurements by observing a calibration source emitting a spectral line close to that of the target emission. Or it could be done simultaneously by observing a calibration source emitting a spectral line different from that of the target emission that is separated into a second channel by placing a dichroic filter in the exit optics."

**Comments:**

(5) line 209: "Several additional criteria were also used to constrain the design*of* the field widened birefringent delay plate."

**Author's response:**

Corrected in the text.

**Author's changes in manuscript:**

It has been corrected in line 277 in the marked-up version.

**Comments:**

(6) line 356, 381: Please put the SNR of 1000 (and the precision of 5 m/s) in perspective with expected atmospheric emission rates and realistic instrument parameters. This is especially important, since this result is used generally later in the paper ("Using these measurements it was shown that wind precisions of < 5 m/s are achievable with the interferometer.") Since the achieved precision is a function of the instrument parameters (etendue, QE, U, V, integration time, etc) and the signal strength, this needs to be expressed more concisely, with these constraints/parameters included.

**Author's response:**

As is described in section 2.2, the expressions for the uncertainty in the wind measurement for the Michelson were originally developed by Ward (1988) and Rochon (2001) and general expressions for the sensitivity of Doppler wind measurements are presented by Kristoffersen et al. (2021). In the ideal case, where four samples are obtained with 90-degree phase steps the expression for the standard deviation, $\sigma_w$, of the wind measurement is

$$\sigma_w = \frac{c\lambda}{2\sqrt{2}\pi(\text{SNR})UVD} \qquad \text{(RC1-6-1)}$$

According to Table 2 and Table 6 in the manuscript, for a field instrument, we assume $UV$ is ~0.5, OPD is 1cm, when measuring the O1D red line (630nm), the wind precision of < 5 m/s is achievable with an SNR of ~1000.

The $O^1D$ red line (630nm) has an expected limb intensity of 30kR. In section 2.2, we give the expression for the signal $S$ at the detector:

$$S = \frac{10^6}{4\pi} E_0 A\Omega\tau\eta t \qquad \text{(RC1-6-2)}$$

The etendue $A\Omega$ for a field instrument is 0.86 cm2sr, QE $\eta$ is 0.9,and $\tau$ is 0.1 when account for the transmittance of the filter and lens system. If the spatial resolution is 100*100, we can get an SNR of ~1000 with an integration time of 55 seconds. However, for the ground-based observation, the intensity of red line is only ~100R. Thus, we need to reduce the spatial resolution and increase integration time. Typically, if the spatial resolution is 10*10, we can get an SNR of ~1000 with an integration time of 165 seconds. This calculation combines the SNR of ~1000, the precision of 5m/s, the expected atmospheric emission rates and realistic instrument parameters.

**Author's changes in manuscript:**

We give the reader a clearer relationship between the instrument parameters and the needed SNR in the revised manuscript. The SNR can be 480 for an optimized ground-based instrument. This is possible since the instrument visibility, U will be higher for the optimized field instrument (U ~ 0.9) than for the prototype which was found to have surface variations that degraded the instrument visibility. We have added some calculations and discussions in Section 4.2, Section 5.2, and the Discussion section to help readers. Please see line 335 to line 343, line 355 to line 367, line 462 to line 466, and line 605 to line 626 in the marked-up version. Some numbers in Table 6 have also been revised to take into account the fact that a portion of the field of view (conservatively estimated at 25%) will be unusable for wind measurements due to lack of quadrature.

**Comments:**

(7) line 440: The authors state: "Comparison of the effectiveness of a field widened birefringent interferometer relative to a field widened Michelson interferometer for the measurement of Doppler winds can be undertaken with respect to the primary instrument design parameters: A, Omega and D." As mentioned above, I think that this is exactly what should be done to motivate this paper, otherwise it is not clear why one would consider this technique for any specific application. Table 6 attempts to do that, but it does not show any compelling improvement over many of the other instruments listed. In addition, the "relative wind precision E" is not a very intuitive metric. Assuming an atmospheric signal strength, one could give the wind precision in meters per second, which would be much more intuitive.

**Author's response:**

The best way to compare the capabilities of different interferometers has been a topic of debate for decades and no agreement exists on the best way to compare instruments. While this metric is not perfect, neither is a direct comparison that assumes a particular atmospheric signal strength. For example, the spatial and temporal sampling of MIADI is chosen to specifically target certain scales of geophysical variability that are not targeted by BIDWIN. Therefore, the spatial binning and measurement cadence also strongly influence the comparison. We sought a simple metric that could be used to compare to several state-of-the-art field widened Michelson interferometers while removing the dependence on the scene radiance and detector parameters (Eq.17).

As mentioned above in the response to the general comment, the primary advantage here is the size and simplicity of the construction of the field widened delay plate. The primary finding

of this paper is that a Doppler wind imaging birefringent interferometer can be constructed with comparable capabilities to existing state of the art field widened Michelson interferometers. Therefore, the compelling improvement is a smaller, simple to construct, instrument with comparable capabilities.

**Author's changes in manuscript:**

The revision is same as the response to the general comment. The Discussion has also been revised.

**Comments:**

(8) line 462: The authors state: "In this case, the birefringent interferometer can achieve a throughput of 0.86 cm2sr and is capable of achieving similar wind errors and yet it has the smallest path difference." It is not clear to me why the "smallest path difference" is an advantage. The path difference should be optimized to the width (temperature) of the emission line.

**Author's response:**

You are correct, the low path difference is not necessarily an advantage. However, the sentence was more meant to highlight that it is interesting that we can achieve low precision wind errors with such a small path difference interferometer by taking advantage of the large SNR provided by the instrument. The statement has been removed from the text.

**Author's changes in manuscript:**

The statement has been removed from the text, please see line 622 in the marked-up version.

**Review # 2**

**Comments:**

A new compact static called birefringent Doppler wind imaging interferometer (BIDWIN) is developed for the purpose of observing upper atmospheric winds using suitably isolated airglow emissions. The data is simulated in the lab using a prototype instrument and the wind is retrieved and compared with wind wheel velocity. The new instrument is validated for use of wind measurements of upper atmosphere. The paper can be accepted considering some questions as follows:

1. In general, this manuscript is too long. The introduction of airglow radiation in section 2.1 can be more concise. It is recommended to incorporate it into the first chapter.

**Author's response:**

We believe that Section 2.1 is an important and distinct part of the paper. First and foremost, it presents the background regarding the various airglow emissions that can potentially be used to measure upper atmospheric winds. Second, it presents the procedure used to examine the relationship between the optical path difference and the sensitivity to Doppler winds for the various emissions. This information will be important for any researcher attempting to reproduce or build upon the current work.

**Author's changes in manuscript:**

We provided references for the projects in Table 1 so the readers can follow-up, please see line 105 to line 109 in the marked-up version.

**Comments:**

2. Line 26 describes FPI and DASH, "primarily in the thermosphere region." This description is not rigorous enough, because UARS/FPI has already realized the middle atmosphere wind field detection (can reach below 40km).

**Author's response:**

The wording "primarily in the thermosphere region" has been removed.

**Author's changes in manuscript:**

The wording "primarily in the thermosphere region" has been removed, please see line 49 to line 50 in the marked-up version.

**Comments:**

3. the Birefringent Doppler Wind imaging Interferometer (BIDWIN) in 2th line of abstract should be moved to the first line.

**Author's response:**

We understand the confusion between the two sentences. Therefore, the two first sentences of the abstract have been revised to say:

   "A new compact static wind imaging interferometer, called the Birefringent Doppler Wind imaging Interferometer (BIDWIN) has been developed for the purpose of observing upper atmospheric winds using suitably isolated airglow emissions. The instrument combines……."

**Author's changes in manuscript:**

The abstract has been revised, please see line 12 to line 21 in the marked-up version.

**Comments:**

4. The influence of stray light is not been considered in the prototype testing. In fact, this is very important that needs to be considered in the availability evaluation of the measurement result. So it is recommended to add.

**Author's response:**

A detailed analysis of stray light is outside the scope of this work. However, one of the advantages of using the four quadrant approach is that the same image of the field of interest appears in each quadrant. As a result, any background light or stray light from the field would appear in each image in the same location and not be modulated. Hence this approach is robust to unpolarized scattered light or stray light without spectral signatures.

   The interferometer would need to be carefully shielded from any stray light which might be incident from the sides and affect one quadrant but careful construction to prevent this would be the same as would be the case for similar instruments. To ensure this point is clear to the

readers we have added the following sentence at the end of section 3.

"While a full analysis of stray light is outside the scope of this work, another important feature of the BIDWIN approach is that each quadrant images the same field. Therefore, the background or scattered light from the field will be unmodulated and appear as a constant offset for the corresponding bins in each quadrant. In this case, the fringe phase will not be affected."

**Author's changes in manuscript:**

We have added the discussion at the end of Section 3, please see line 248 to line 250 in the marked-up version.

**Comments:**

5. In the evaluation of wind velocity measurement accuracy, the light source used by the author is He-Ne laser, but the intensity of the laser is several orders of magnitude larger than the intensity of airglow. Please explain whether the testing results are convincing.

**Author's response:**

Thanks for your comments. As is described in section 2.2, the expressions for the uncertainty in the wind measurement for the Michelson were originally developed by Ward (1988) and Rochon (2001) and general expressions for the sensitivity of Doppler wind measurements are presented by Kristoffersen et al. (2021). In the ideal case, where four samples are obtained with 90-degree phase steps the expression for the standard deviation, $\sigma_w$, of the wind measurement is

$$\sigma_w = \frac{c\lambda}{2\sqrt{2}\pi(\text{SNR})UVD} \tag{RC2-5-1}$$

As is discussed in the section 5 of the manuscript, the wind precision of 5 m/s is achievable with an SNR of ~1000 (laser test results). Although the intensity of the laser is several orders of magnitude larger than the intensity of airglow, we will show here the SNR of ~1000 is achievable when observing airglow and the lab test results are convincing.

In section 2.2, we give the expression for the signal $S$ at the detector:

$$S = \frac{10^6}{4\pi} E_0 A\Omega\tau\eta t \tag{RC2-5-2}$$

According to Table 2 and Table 6 in the manuscript, the O$^1$D red line (630nm) has an expected limb intensity of 30kR. The $A\Omega$ for a field instrument is 0.86 cm2sr, $\tau$ is 0.1 when account for the transmittance of the filter and lens system, $\eta$ is 0.9. If the spatial resolution is 100*100, we can get an SNR of ~1000 with an integration time of 55 seconds. That is to say, we can get a similar SNR whether observing laser or airglow, so the lab test results are convincing.

**Author's changes in manuscript:**

We give reader a clearer relationship between the wind precision and the needed SNR in the revised manuscript. The SNR can be 480 for an optimized ground-based instrument. This is possible since the instrument visibility, U will be higher for the optimized field instrument (U ~ 0.9) than for the prototype which was found to have surface variations that degraded the instrument visibility. We have added some calculations and discussions in Section 4.2, Section 5.2, and Discussion to help readers. Please see line 335 to line 343, line 355 to line 367, line

462 to line 466, and line 605 to line 626 in the marked-up version. Some numbers in Table 6 have also been revised.

**Comments:**

6. Figure 13 just shows a result of wind drift with time, how about the result of wind drift with temperature?

**Author's response:**

Figure 13 shows the thermal drift of the instrument due to the thermal dependence of the optical path difference. Since the temperature in the enclosure is changing as a function of time, the optical path difference is changing as a function of time. So the drift is actually function of the changing temperature.

**Author's changes in manuscript:**

We have added the discussions on thermal drift of BIDWIN to Section 3, Section 4.2, and Discussion. Please see line 239 to line 247, line 286 to line 287 and line 576 to line 580 in the marked-up version.

**Other changes in manuscript:** Some other minor changes have been made in the revised manuscript for typos and grammatical errors. Please see those changes in the marked-up version.